Cytological analysis of tracheal wash and bronchoalveolar lavage fluid in health and respiratory disease in dromedary camels

Shawaf Turke tshawaf@kfu.edu.sa 1
Almubarak Abdullah 2
Alhumam Naser 2
Almathen Faisal 3 4
Hussen Jamal 2
1 Department of Clinical Sciences, College of Veterinary Medicine, King Faisal University , Al-Ahsa , Saudi Arabia
2 Department of Microbiology, College of Veterinary Medicine, King Faisal University , Al-Ahsa , Saudi Arabia
3 Department of Public Health, College of Veterinary Medicine, King Faisal University , Al-Ahsa , Saudi Arabia
4 The Camel Research Center, King Faisal University , Al-Ahsa , Saudi Arabia
Aly Sharif
Electronic publication date: 2021 Jul 2
Publication date: 2021
Volume: 9
Electronic Location ID: e11723
Received 2020 Nov 11; Accepted 2021 Jun 14
Copyright: ©2021 Shawaf et al.
Copyright year: 2021
Copyright holder: Shawaf et al.
License: This is an open access article distributed under the terms of the Creative Commons Attribution License, which permits unrestricted use, distribution, reproduction and adaptation in any medium and for any purpose provided that it is properly attributed. For attribution, the original author(s), title, publication source (PeerJ) and either DOI or URL of the article must be cited.
License URL: https://creativecommons.org/licenses/by/4.0/

Keywords: Bronchoalveolar lavage, Camel, Cytology, Respiratory, Tracheal wash

Funding: Deanship of Scientific Research at King Faisal University Al-Ahsa, Saudi Arabia, under Research Group Support Track 1811001 This work was supported by the Deanship of Scientific Research at King Faisal University, Al-Ahsa, Saudi Arabia, under Research Group Support Track (Grant No. 1811001). The funders had no role in study design, data collection and analysis, decision to publish, or preparation of the manuscript.

==============================
Background

Tracheal wash (TW) and bronchoalveolar lavage (BAL) have proven to be useful tools for the identification of disease-associated changes in the respiratory tract in human and different animal species. In the dromedary camel, little is known about cytological analysis of TW and BAL in health and disease. The aim of the present study was to evaluate the cytological composition of TW and BAL in health and respiratory disease in dromedary camels.

Methods

TW and BAL samples were collected from dromedary camels and cytological analysis was performed by microscopic examination of prepared smears. Camels with clinical respiratory disease (n = 18) were compared with apparently healthy (control) camels (n = 9).

Results

In the apparently healthy camels, differential cytological analysis of TW samples identified macrophages and neutrophils as the main cell populations with lesser proportions of lymphocytes and epithelial cells and very rare abundance of eosinophils and mast cells. In the TW of camels with respiratory disease, neutrophils were the most abundant cells followed by macrophages and lymphocytes. In the BAL of healthy camels, macrophages represented the main cell type followed by lymphocytes and neutrophils. In respiratory-diseased camels, BAL samples contained higher percentages of neutrophils with reduced percentages of macrophages and lymphocytes in comparison to camels from the control group. Collectively, the results of the current study revealed higher abundance of neutrophils in the TW and BAL from dromedary camels than many other veterinary species. The cytological patterns of TW and BAL from camels with respiratory diseases were characterized by increased proportion of neutrophils and decreased proportion of macrophages in comparison to healthy camels. The proportion of lymphocytes was also decreased in TW samples from diseased camels.

Introduction

The dromedary camel (Camelus dromedarius) belongs to the most important domestic animal species adapted to dry and hot regions in Asia and Africa (Kebedi, 2010). The camel world population is around 30 million heads (Faye, 2015). Respiratory diseases are among the main factors responsible for high mortality rates in dromedary camels (Al-Ruwaili, Khalil & Selim, 2012); Bakhesh (Alhendi, 2000; Gebru et al., 2018; Li et al., 2017; Scaglione et al., 2017). Among respiratory infectious diseases, viral and bacterial infections are usually correlated with most losses (Schwartz, 1992). The main viral respiratory pathogens in camels include adenovirus, parainfluenza 3 virus (Kebedi, 2010), morbilivirus (Al-Rawashdeh et al., 2000), and MERS Coronavirus, which is among the emerging public health hazards (Memish et al., 2013). Bacterial infections of the respiratory tract of camels have been shown to be mainly caused by Manhaemiya hemolytica, Corynebacterium spp., Pasteurella spp., and Arcanobacterium pyogenes (Wareth et al., 2014; Wernery , 2002).

The early diagnosis of respiratory disorders is a key factor for choosing specific treatment to prevent disease progression and chronic lung disease (Intisar et al., 2009; Intisar et al., 2010a; Intisar et al., 2010b; Kebedi, 2010). Tracheal wash (TW) and bronchoalveolar lavage (BAL) are valuable tools widely used for the investigation of the respiratory tract in human (Rose & Knox, 2007), horses (Orard et al., 2016), cattle (Pringle et al., 1988; Abutarbush et al., 2019; Kokotovic, Friis & Ahrens, 2007), sheep (Katsoulos et al., 2009), pigs (Weissenbacher-Lang et al., 2016), alpaca (Pacheco, Mazan & Hoffman, 2012), dogs (Creevy, 2009; Zhu, Johnson & Vernau, 2015), and cats (Lin et al., 2015). TW and BAL samples are valuable mirrors of different areas of the airways and can be used in combination with other methods for the diagnosis of lower respiratory tract inflammation (Malikides et al., 2003). Being able to explore large areas of the lower respiratory tract (Hoffman, 2008), BAL is widely used as a source for specimens for cytological, microbiological and immunological investigation of the respiratory tract (Couetil et al., 2007; Hoffman, 2008; Hoffman, Mazan & Ellenberg, 1998; Rose & Knox, 2007). BAL and TW samples are also useful in giving an insight on the severity and stage of inflammatory reactions in the lung and in the detection of subclinical pulmonary diseases (Caldow, 2001). Species-specific differences have been reported regarding the selection of the appropriate technique for sampling the respiratory secretions. In horses, TW is preferred for diagnosis of respiratory infections, whereas BAL is the best choice for the diagnosis of non-infectious pulmonary diseases (Davis & Sheats, 2019). However, similar diagnostic values have been reported for BAL and TW for the diagnosis of common bacterial and viral respiratory pathogens in cattle (Doyle et al., 2017). Although respiratory cytology may rapidly provide a presumptive diagnosis of infection, identification of the etiologic agent requires bacterial or viral culture or PCR (Jocelyn et al., 2018).

Although some studies reported the cytological composition of TW samples in camels (Habasha & Hussain, 2014; Habasha & Hussain, 2016) , there have been no studies conducted on the comparative analysis of TW and BAL cytology in dromedary camels. The aim of the current study, therefore, was to comparatively analyze the TW and BAL cytology in healthy and respiratory-diseased dromedary camels.

Materials & Methods

Ethical approval

All experimental procedures used in this study were approved by the Ethics Committee at King Faisal University, Saudi Arabia (Permission number KFU-REC / 2019 –10 - 01). All applicable international, national, and institutional (King Faisal University) guidelines for the care and use of animals were followed.

Animals and experimental design

Twenty-seven camels (Camelus dromedaries) of different ages, sex, and breeds were involved in this study. The control group included nine healthy camels (five males and four females with a median age ± SEM of 10.3  ± 5.5 years and a median weight ± SEM of 390 ± 125 kg) selected from animals maintained at the Camel Research Center at the King Faisal University, Al-Ahsa, Saudi Arabia. The diseased animal group included eighteen camels (seven males and eleven females with a median age  ± SEM of 12  ± 6 years and a median weight ± SEM of 405 ± 185 kg) randomly selected from respiratory-diseased camels brought to the Veterinary Teaching Hospital, College of Veterinary Medicine, King Faisal University. Control camels were selected based on clinical scoring, physical, and laboratory evaluation (no fever or signs of abnormal respiratory signs such as cough, nasal discharge, dyspnea or abnormal lung sounds as well as normal white blood cell counts and biochemistry panel).

Clinical examination

For all animals, detailed history and clinical examination signs (dyspnea, cough and nasal discharge, lung sounds, and rate of breathing) were recorded. For the clinical evaluation of control and respiratory diseased camels, a scoring system designed for healthy and respiratory diseased horses (Ohnesorge & Deegen, 1998) was adapted with a modification by considering the animal rectal temperature in the scoring procedure (Love et al., 2014) (Table 1). Clinical score points were recorded by two persons. However, the data could not be taken blindly due to the difficulty of taking the case history twice in the hospital. Camels with a clinical score less than three points were considered healthy while camels with three or higher score points were considered diseased.

Table 1 Detailed scoring system points for respiratory-affected camels.

Parameter	Clinical findings	
Cough	
0	None	
1	Rare	
2	Daily repeated	
3	Spontaneous during clinical examination	
Dyspnoea	
0	No difficulty in respiration	
1	Mild visible increase of the abdominal movement at the end of expiratory phase.	
2	A clear abdominal movement during expiration.	
3	A difficult abdominal breathing with wide nostrils.	
4	High degree of respiratory distress	
Lung auscultation	
0	Physiological (vesicular inspiration)	
1	Mild degree of sharpness of inspiratory sounds.	
2	Exaggerated inspiratory sounds and quite expiratory sounds.	
3	Roughness in both expiratory and tracheal rattle sounds.	
4	“Wheezes, Crackles” and or rattle sounds over the lung area.	
Rectal temperature	
0	<36.5	
1	36.7–39	
2	39.1–40	
3	>40.1	

Bronchoscopy and collection of TW and BAL samples

For bronchoscopy, camels were positioned in sternal recumbency position. Due to the narrow nasal passage in camels, especially in the Omani breed, and the lack of camel-specific endoscope and BAL catheters, endoscope and BAL catheters were inserted via the oral cavity. After the intravenous injection of xylazine 2% (0.1 mg/kg bodyweight; Rompun, Bayer Health Care) (Shawaf et al., 2017), a special mouth gag especially designed for camels in the Veterinary Teaching Hospital (Fig. 1) was placed to keep the mouth open protecting and allowing easy passage of the endoscope. The sedation was maintained throughout the whole endoscopic TW and BAL collection procedures by intravenous injecting xylazine (0.05 mg/kg). A flexible 3.2 m long, 12 mm tip diameter bronchoscope (EVIS Olympus, OLYMPUS AUSTRIA GmbH., Vienna), supported with an insufflation system, light source, and irrigation system was introduced into the oral cavity. When reached the pharynx, the endoscope was inserted via the rima glottidis into the trachea. A tiny sterile plastic catheter (EQUIVET; 2, 3 mm ×350 cm) was passed into the endoscope’s working channel and TW was done by the injection of 10–15 mL sterile normal saline into the last part of the trachea followed by immediate aspirating to recover TW fluid. The retrieved fluid was collected into plain tubes for cytological analysis (within 15 min of collection). BAL was performed immediately after the endoscopy and TW procedure. To decrease coughing during BAL, 20 −40 mL of 1% lidocaine was infused as local anesthesia into the lower airway. A catheter (EQUIVET B.A.L. catheter 240 cm, KRUUSE, Denmark) was passed through the speculum of mouth gag into the oral cavity until the pharynx and then advanced into the larynx, trachea, and bronchi until reaching a slight resistance (Fig. 2). As we faced difficulties to introduce the BAL catheter blind into the larynx, we used an endoscope for guiding the BAL catheter into the larynx. As soon as the tube was wedged into the bronchus, the cuff was then gently inflated using 10–20 ml of air to prevent the backflow of infused fluid. Five syringes (each of 50 ml) of sterile isotonic saline were placed in a water-bath to warm up to approximately 37 °C and were then instilled via BAL catheter. BAL was aspirated immediately after injection and the samples were immediately positioned on ice and submitted to the lab within 30 min of collection. Samples were considered acceptable when they contained a foamy surfactant layer. BAL samples were used for cytological analysis within one hour after sample collection.

Figure 1 Placement the endoscope to preform the endoscopic examination and TW sample collection from dromedary camel.

Figure 2 B. ronchoalveolar lavage collection using BAL catheter from dromedary camel.

Cytological analysis of TW and BAL samples

The total cell count of BAL samples was estimated by taking a small volume from the well mixed original fluid in a Bürker’s counting hemocytomer (Laporoptic, Germany) after filtering the sample through a gauze. For differential cell counting, the TW and BAL samples were centrifuged at 1,500 rpm for 15 min and the supernatant was discarded. Microscopic smears were prepared from the cell pellet and air dried smears were stained using the Diff Quick staining kit (Hemal Stain Co. Inc., Danbury, CT). The cytological analysis was performed by a specialist in clinical pathology, who was blinded to the camel health status. Using a magnification of 1, 000 × and a standardized counting protocol (De Brauwer et al., 2000; De Brauwer et al., 2002), the percentages of macrophages, lymphocytes, neutrophil, mast cells, eosinophils, and epithelial cells were calculated after counting a minimum of 400 cells.

Statistical analysis

Using the statistical software Graph Pad Prism 5, differences between the means were analyzed using the one-way analysis of variance (ANOVA) in combination with the Bonferroni post test for multi-comparison analysis. Normal distribution was evaluated by D’Agostino & Pearson omnibus normality test. The differences between the groups were considered significant if the P-value was less than 0.05.

Results

For the cytological analysis of tracheal wash (TW) and bronchoalveolar lavage (BAL) in healthy and respiratory diseased dromedary camels, a scoring system was used to group camels into a healthy control group (n = 9) and a respiratory diseased group (n = 18). All examination parameters and scoring points were within the normal ranges in the animals from the control group with no abnormal respiratory sounds or any signs of respiratory disorders or infections (clinical score <3 points with a mean ± SEM of 0.88 ± 0.75). The diseased group showed varying signs of respiratory signs like cough, nasal discharges, and abnormal respiratory sounds (clinical score ≥ 3 points with a mean ±SEM of 7.9 ±3.4). Endoscopic visualization of the mucosa lining the trachea and bronchi also revealed no abnormal mucus accumulation in healthy animals, while there was varying degrees of mucus accumulation and narrowing in the airways of diseased camels. From the total instilled 250 mL fluids, the retrieved fluids were 155 ± 35.2 mL, representing 62 ± 14.8% of the total instilled fluids. No differences in the volume of the retrieved fluid were observed between healthy and diseased camels.

Due to specific anatomical and physiological characteristics of the pharynx cavity and the long soft palate (Dulla) in the camel, we needed to use visual endoscopy in eight cases to introduce the BAL catheter into the larynx.

Identification of different cell types in camel TW and BAL samples

In both TW and BAL samples, camel alveolar macrophages were slightly variable in size with abundant vacuolated cytoplasm and irregular cell margins (Figs. 3A and 3E). Macrophages cytoplasmic vacuoles occasionally contained cellular debris (Fig. 3E). BAL neutrophils were normally segmented and non-degenerative, which is similar to blood neutrophils, whereas TW neutrophils were segmented and degenerated (Fig. 3A). Lymphocytes were characterized by small round central to eccentric nuclei with dense clumped chromatin and scant amounts of blue cytoplasm with smooth margins (Fig. 3E). Eosinophils were identified based on their uniformly sized small red orange cytoplasmic granules (Fig. 3C). Epithelial cells were found as ciliated columnar cells with basally located nucleus. Some epithelial cells showed loss of cilia, which were seen in the background of the slides (Fig. 3A and 3F). In general, TW cells were frequently degenerated and more difficult to be differentiated in comparison to BAL cells. In some TW slides, contamination with bacteria, saliva, food material, red blood, or oral squamous epithelial cells was observed (Fig. 3D).

Figure 3 The identification of different cell types in TW and BAL samples.

Cytological slides were prepared from TW and BAL samples, stained with the Diff Quick stain, and examined microscopically: (A) A TW slide showing different cell types, including neutrophils (N) and epithelia cell (Ep) with a magnification of 1,000×; (B) a TW slide from affected camel showing increased quantity of degenerated neutrophils with a magnification of 200×; (C) a TW slide from a camel with respiratory disease showing degenerated neutrophils. This field also shows an eosinophil cell with granules within the cytoplasm with a magnification of 1,000×; (D) a TW slide from a camel with respiratory disease showing oral epithelial cells (O Ep), bacteria (B), red blood cells (RBC), and separated cilia (C). Stain Diff Quick stain; magnification, 1,000×; (E) a BAL slide prepared from a healthy camel showing macrophages (M) and Lymphocytes (L); (F) a BAL slide from a camel with respiratory disease showing increased numbers of neutrophils. In addition, macrophages (M), alveolar macrophage (AM), lymphocytes (L), and epithelia cell (Ep) can be identified in the field; magnification, 1,000×.

Cellular composition of TW and BAL samples from healthy and diseased camels

BAL samples from diseased camels contained significantly more cells (824 ± 401.1 cells/µL) than BAL samples from the control group (200.4 ± 39.2 cells/µL) (Table 2). Due to high mucus accumulation in the samples, we found difficulties in the estimation of total cell counts in the TW samples.

The differential cell counts of the TW fluids in healthy camels consisted primarily of macrophages (51.6 ± 10.2%) and neutrophils (27.3 ± 7.2%) with lesser frequency of lymphocytes (8.2 ± 1.3%), epithelial cells (8.8 ± 5.1%), eosinophils (1.7 ± 0.4%), and mast cells (1 ± 0.4%). In the TW samples from diseased camels, neutrophils were the most abundant cells (73.3 ± 7.4%) followed by macrophages (22.4 ± 7.6%) with lower percentages of lymphocytes (3.2 ± 1.2%), mast cells (1.3 ± 0.1%), epithelial cells (1.2 ± 0.3%), and eosinophils (0.5 ± 0.4%) (Figs. 4A–4F).

Table 2 Total and differential cell counts (Mean ± SEM and range) in TW and BAL fluids from 27 camels (9 healthy and 18 affected).

	Tracheal Wash (TW)	Bronchoalveolar Lavage Fluid (BAL)	
	Healthy	Affected	Healthy	Affected	
	Mean ± SEM	Range	Mean  ± SEM	Range	Mean  ± SEM	Range	Mean  ± SEM	Range	
Total cell count	–	–	–	–	200.4  ± 39.22	60–440	824  ± 401.1**	260–2400	
Macrophages %	51.60  ± 10.25	22–72	22.46  ± 7.69	6–45	60.01  ± 3.11	46–75	52.66  ± 4.24	37–66	
Lymphocytes %	8.20  ± 1.3	4–11	3.2  ± 1.2	1–7	23.74  ± 1.93	14–35	15.80  ± 6.11	11–33	
Neutrophils %	27.34  ± 7.21	12–54	73.30  ± 7.42	51–88	7.95  ± 1.80	2–19	24.68  ± 4.67	16–39	
Mast cells %	1  ± 0.44	0–2	1.36  ± 0.13	0–3	0.58  ± 0.21	0–2	1.08  ± 0.72	0–3	
Eosonophils%	1.72  ± 0.42	1–3	0.48  ± 0.38	0–2	1.6  ± 0.5	0–4	2.14  ± 0.92	0–6	
Epithelia cells %	8.8  ± 5.08	2–29	1.2  ± 0.37	0–2	5.45  ± 1.2	1–11	5.18  ± 1.16	1–8	
Notes.

** P < 0.01.

The differential cell counts of the BAL in healthy camels revealed the dominance of macrophages (60.0 ± 3.1%) and lymphocytes (23.7 ± 1.9%) with low proportions of neutrophils (7.9 ± 1.8%) and epithelial cells (5.4 ± 1.2%) and rarely seen mast cells (0.6 ± 0.2%) and eosinophils (1.6 ± 0.5%). BAL samples from diseased camels showed reduced percentage of macrophages (52.6 ± 4.2%) and lymphocytes (15.8 ±6.1%) but increased percentage of neutrophils (24.7 ± 4.7%) when compared to healthy animals. However, the percentage of epithelial cells (5.2 ± 1.2%) was similar in healthy and diseased camels (Figs. 4A–4F). Based on their rectal temperature, the studied animals were classified into animals with low temperature (<39) and animals with high (>39.1) temperature. BAL samples from the animal group with high rectal temperature contained significantly more cells (1653 ±235.6 cells/µL) than BAL samples from the normal group (587 ±115.7 cells/µL) (Table 3). For both TW and BAL samples, the percentage of neutrophils was higher in the camels with high rectal temperature . In addition, there was a decrease in the percentage of macrophage in TW and BAL samples in animals with high rectal temperature compared to animals with normal rectal temperature (Table 3).

Discussion

In the current study, the cytological composition of tracheal wash (TW) and bronchoalveolar lavage (BAL) samples were comparatively analyzed in apparently healthy camels and camels with clinical respiratory diseases. The cytological patterns of TW and BAL from camels with respiratory diseases were characterized by increased proportion of neutrophils and decreased proportion of macrophages in comparison to healthy camels. The proportion of lymphocytes was also decreased in TW samples from diseased camels, when compared to healthy camels.

The detecting of oral epithelial cells with bacterial contamination in TW slides may affect the cytological analysis and argues against using TW for bacterial examination (Smith, 2019). In the horse, other sampling techniques have been suggested for obtaining uncontaminated lower airway secretions for bacterial culture. This includes the transtracheal aspiration after the insertion of a sterile flushing tube through a tracheal cannula between tracheal rings below the larynx. In addition, the protected aspiration catheter technique using a guarded sterile tube, due to its reduced complication risk, has been suggested as an alternative to the transtracheal aspiration for the isolation of equine pulmonary bacteria (Darien et al., 1990). However, comparative studies are required to determine the most optimal sampling technique for the collection of camel lower airway fluids for bacterial culture.

Figure 4 The differential composition of TW and BAL samples.

Cytological slides were prepared from TW and BAL samples, stained with the Diff Quick stain, and examined microscopically. The percentage of macrophage cells (A), lymphocytes (B), neutrophils (C), eosinophils (D), mast cells (E), and epithelial cells (F) were estimated in the total cellular content of BAL and TW samples collected from healthy camels and camels with respiratory diseases. Date is presented as mean ±standard error of the mean (SEM). There were considered to be differences in mean values when there was a P-value of less than 0.05.

TW contamination with saliva might have resulted from oral saliva coming down from the oral cavity into the airways during endoscopy. As shown in Fig. 3D, the presence of some red blood cells in TW could be due to minimal bleeding during sample collection (Hughes et al., 2003). Our results are also in agreement with Walker et al. (2006), who observed that BAL cells are better preserved and usually easier to be identified than TW cells.

Table 3 Total and differential cell counts (Mean ± SEM and range) in TW and BAL fluids from 27 camels regarding rectal temperature (six with normal (<39) rectal temperature and 22 with high (>39.1) rectal temperature) healthy and 18 affected).

	Tracheal Wash (TW)	Bronchoalveolar Lavage Fluid (BAL)	
	Normal rectal temperature (<39)	High rectal temperature (>39,1)	Normal rectal temperature (<39)	High rectal temperature (>39,1)	
	Mean ± SEM	Range	Mean  ± SEM	Range	Mean  ± SEM	Range	Mean  ± SEM	Range	
Total cell count	–	–	–	–	587  ± 115.7	60–2250	1653 ± 235.6*	815–2400	
Macrophages %	36.24  ± 4.55	6–72	17.43  ± 3.99*	6–31	56.71  ± 2.06	37–75	48.58  ± 4.34	37–65	
Lymphocytes %	5.1  ± 0.82	1–11	4.17  ± 0.94	1–7	19.93  ± 2.02	1–35	11.67  ± 5.19*	1–35	
Neutrophils %	51.22  ± 5.9	12–88	78.83  ± 3.89*	62–88	14.76  ± 1.99	2–36	31.42  ± 4.33*	17–49	
Mast cells %	1.37  ± 0.21	0–3	1.1  ± 0.48	0–3	1.17  ± 0.24	0–3	1.33  ± 0.61	0–3	
Eosonophils%	1.1  ± 0.22	0–3	0.47  ± 0.32*	0–2	1.62  ± 0.32	0–5	2.34  ± 1.05*	0–6	
Epithelia cells %	4.48  ± 1.44	0–29	1.33  ± 0.33*	0–2	5.48  ± 0.61	1–11	4.67  ± 1.28	1–8	
Notes.

* P < 0.05.

Due to limited information about cytological values in camels, the results from the current work were compared with data reported for other species. In our study, the total cell count of BAL in healthy camels was similar to results reported for the bovine BAL cytology (Abutarbush et al., 2019). Similar to other species, including cattle, horses, and donkeys (Abutarbush et al., 2019; Hoffman, 2008; Rossi et al., 2018; Shawaf, 2019), respiratory diseases in camels were associated with increased total cell count in the BAL fluid. For cattle with respiratory diseases, however, greater increase in the total BAL cell count was reported (Thirunavukkarasu et al., 2005).

Similar to their distribution in healthy horses (Malikides et al., 2003), neutrophils were found in higher proportions in TW than in BAL fluids from healthy camels. The increased fraction of neutrophils in TW and BAL fluids from camels with respiratory diseases is also in line with findings in respiratory diseased horses (Rossi et al., 2018), donkeys (Shawaf, 2019), cattle (Kokotovic, Friis & Ahrens, 2007), and alpaca (Pacheco, Mazan & Hoffman, 2012).

In the present study, the decreased fraction of macrophages in TW and BAL from diseased camels is in line with reports from other species, including human (Rose & Knox, 2007), horses (Rossi et al., 2018), cattle (Kokotovic, Friis & Ahrens, 2007), and alpaca (Pacheco, Mazan & Hoffman, 2012). The lower frequency of macrophages in diseased animals may be a result of the increased accumulation of neutrophils in the respiratory secretions. Further studies are needed to see, whether this change in macrophages count is also associated with modifications in their functional type.

Although the proportion of lymphocytes in the BAL fluids from healthy camels is comparable with their percentage in equine BAL fluids, healthy camel TW contained only a minor population of lymphocytes, which is in contrast to the equine system, where TW lymphocytes are also the second dominant population after macrophages (Richard et al., 2010). In contrast to finding in the horse, where changes in the percentage of BAL and TW lymphocytes were of lower relevance for the diagnosis of equine respiratory disorders (Hoffman, Mazan & Ellenberg, 1998; Rossi et al., 2018; Shawaf, 2019), we found significantly less lymphocytes in TW of respiratory diseased camels. Whether these differences in the cellular composition of respiratory secretions rely on species-specific defense mechanisms in the respiratory tract, further comparative studies are required. Furthermore, as lymphocytes are a heterogeneous cell population, it still to be investigated, whether selective lymphocyte subsets like helper or cytotoxic T cells, B cells, or NK cells, were especially affected by this decrease.

The low frequency of eosinophils (Abutarbush et al., 2019; Hughes et al., 2003) and mast cells (Hughes et al., 2003; Leclere et al., 2006; Malikides et al., 2003) in BAL and TW from healthy camels of the present study is in agreement with results from other species. Although the diagnostic value of mast cells in BAL and TW are not fully studied (Rossi et al., 2018), we found significantly more mast cells in the BAL from diseased camel, which is similar to results reported for diseased horses (Leclere et al., 2006).

As normal cells lining the trachea, epithelial cells are present in high numbers in a normal tracheal wash (Zhu, Johnson & Vernau, 2015) but only in low numbers in BAL samples (Hoffman, 2008). The higher frequency of epithelial cells in TW from healthy than respiratory diseased camels is in contrast to reports from previous studies conducted in horses (Wysocka & Klucinski, 2015). Interestingly, we found in some TW samples from diseased camels separated epithelial cell cilia, which could have originated from the inflamed airways (McCauley et al., 1998; Simet et al., 2010; Sisson et al., 1994). In human, ciliocytophthoria of nasal epithelial cells has been reported after viral infections (Gelardi & Ciprandi, 2019). Investigating the clinical significance of the observed ciliated epithelial cells and its association with viral pathogens, however, requires further studies.

Finally, one of the limitations of the present study is the cell identification method. Although it is widely used for cytological analysis in several species (Jackson et al., 2013), staining with Diffquick may fail to reliably identify all cell types (Leclere et al., 2006). Specifically, the estimated frequency of mast cells in the BAL samples may have been affected by Diffquick staining. However, mast cells represent only a minor fraction within BAL cells, which argues against a significant impact of the staining method on the results of cytological analysis in the present study. The identification of mast cells in other species relies on histochemical stains for their heparin, glycosaminoglycans, or esterase. In addition, different antibodies have been used to identify mast cells in human (Ribatti, 2018). Therefore, the identification of cross-reactive antibodies to camel mast cell markers would enable their confirmed immunophenotypic identification in camels. In addition, the identification of functional cell subtypes, including pro-inflammatory (M1) and anti-inflammatory (M2) macrophages, would help in better understanding and interpretation of cytological findings. For this, a combination of cell labeling with monoclonal antibodies to selected cell surface markers with flow cytometric analysis may be a good alternative to Diffquick staining.

Conclusion

The present study provides the first report on the comparative analysis of TW and BAL cytology in dromedary camels. Dromedary camels show higher abundance of neutrophils in their TW and BAL than many other veterinary species. The cytological patterns of TW and BAL from camels with respiratory diseases are characterized by increased proportion of neutrophils and decreased proportion of macrophages in comparison to healthy camels. The proportion of lymphocytes was decreased only in TW samples from diseased camels. Collectively, BAL and TW represent valuable techniques for detailed investigation of disease-associated cytological changes in the respiratory tract of camels.

Supplemental Information

Supplemental Information 1 The number of studied camels with data of age, sex, breed and respiratory clinical score points

Click here for additional data file.

Supplemental Information 2 Clinical score in details

Click here for additional data file.

Supplemental Information 3 Cytological counts of tracheal wash and bronchoalveolar lavage

Click here for additional data file.

Additional Information and Declarations

Competing Interests

Author Contributions

Ethics

Data Availability

The authors declare there are no competing interests.

Turke Shawaf and Jamal Hussen conceived and designed the experiments, performed the experiments, analyzed the data, prepared figures and/or tables, authored or reviewed drafts of the paper, and approved the final draft.

Abdullah Almubarak and Naser Alhumam conceived and designed the experiments, analyzed the data, authored or reviewed drafts of the paper, and approved the final draft.

Faisal Almathen performed the experiments, analyzed the data, prepared figures and/or tables, authored or reviewed drafts of the paper, and approved the final draft.

The following information was supplied relating to ethical approvals (i.e., approving body and any reference numbers):

All experimental procedures used in this study were approved. The King Faisal University granted ethical approval to carry out the study within its facilities (Permission number KFU-REC / 2019 –10 - 01).

The following information was supplied regarding data availability:

The data are available in the Supplemental Files.

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
