# Peer review of "Cytological analysis of tracheal wash and bronchoalveolar lavage fluid in health and respiratory disease in dromedary camels"

_PeerJ, doi:10.7717/peerj.11723_

## Round 0.1 · original submission · Major Revisions

Experts have reviewed your manuscript and despite finding merits to your study they have shared a number of major concerns that preclude the acceptance of the current version. I invite you to respond to their comments line by line, indicating either the changes made or your rebuttal otherwise.

·

Basic reporting

The literature sorucesd can be reduced substantially without loss of support. As well, removal of some papewrs not peer reviewed work and some that are inappropriate refences is needed. See comments to author

Experimental design

The research question is important and of interest. Additional details are required within the material and methods (see comments to the authors) are needed. As well, the authors need to provide more specific disease/ diagnosis aspects of the nonhealthy cohort, this group as currently presented is inclear as to how to relate to diffierent defined diangoses (ie, some viral infections? And aspiration phemonia?? would differ greatly.
Statistics, suggest a post hoc adjustment of p values, and providendata with confidence intervals

Validity of the findings

The the cofindings appear valid, with the caveat that the interpretation of changes in proportion of cell type need to include consideration of codependance of each cell type with others.

Additional comments

This work of describing BAL and TW cytology in normal and diseased camels is of great interest to the veterinary community, in adding to the range of species these two diagnostic tests are used for study of respiratory disease.
The paper is generally well written and aims are clear. The figures and tables included are relevant, and the supplementary information (excel files) important for transparency of the overall work. The title may be suited to slight modification, to state that this work was performed in healthy AND diseased camels. Ie:
Cytological analysis of tracheal wash and bronchoalveolar lavage fluid in health and respiratory disease in dromedary camels

There are however some aspects of the material and methods that could benefit from additional details (see comments below).
As well, with use of references the number could be reduced considerably (likely fewer than half the number would suffice, and references non-peer reviewed works (Vet Clin North America, Congress proceedings, and non-searchable papers (ie “ Shawaf t. 2011. Evaluation of Equine Bronchoalveolar Lavage Fluid (BALF) and Tracheal Aspirate Sample (TAs) in Normal and Recurrent Airway Obstruction (RAO) Horses. Berlin: mensch und buch verlag” does not come up on pubmed or google) should be removed.

Specific comments: Material and Methods.
My first question is the inclusion of ”diseased “ cohort. While the supplementary information provides details of clinical score this group is in all likelihood very heterogeneous. The paper would benefit and be of great interest to the reader by inclusion include a form of clinical diagnosis for the diseased. This would allow subclassification. For example, were some animals affected by bacterial pneumonia? On the other hand, were there issues of environmental problems – noninfectious airway inflammation such as occurs in asthma in horses? It appears that the clinical scores varied widely within the diseased group. Thus, it would be helpful to both provide clinical diagnoses, and conduct a subclassification of this larger cohort (ie, those with clinical scores 3-5, those 5 or greater? Those with and without fever, which is most simple and defendable to subgroup.) Once the diseased cohort has this subclassification, re-analyses of the TW and BAL results may provide additional highly relevant diagnostic information to the reader.
Regarding the procedure of conducting cytology, the authors should state who (one person, several different persons? trained clinical pathologist?) performed the cell differentials. As well, were those conducting cytology blinded to the health status of the camel?
Moreover, in use simply of Diffquick staining the authors should provide rational, if this has any possible weaknesses in how one would interpret the results; cell distribution? Were there alternative methods that could have been used? For example, classification of cells, in particular mast cells, other stains can be of an advantage in identifying cells that would be missed with Diffquick. As well, differentiation of macrophages from lymphocytes (large granular lymphocytes versus smaller poorly vacuolated alveolar macrophages) gains from selective cells staining or immunologic tagging (ie, see the reference Osbaldiston and Sullivan Cytochemical demonstration of esterases in peripheral blood leukocytes. Am J Vet Res 1978… the method which was used for classification in calf BAL- Pringle …. Valli . Can J Vet Res 1988)- undoubtedly cell identification methods have improved even more in the past several decades. Please provide comment, either in material and methods or, perhaps more fitting, in discussion, of the influence your choice of the specific methodology might have influenced your overall results (and conclusions).
Regarding BAL, in my own experience (horses), some individuals with severe asthma will have a very low volume of fluid recovered on the BAL procedure. While the authors provide total cell counts (per volume) do they have data on volume recovered? And if so, it would held the evidence base of the work to provide the information; if the volumes between healthy and disease did not differ the results presented would not be markedly altered, whereas significantly different volume recovered on BAL should be included in the analyses alongside the total cell numbers per volume. As well, the descriptive statistics of results of cytology differentials and counts will suffice with one decimal point in reporting
Regarding statistics, as very many comparisons were being made with the ANOVA, please consider applying a post hoc adjustment/correction of P values; Bonferroni or the like, as just by chance with the multiple comparisons some with be found to P< 0.05. It would also be suitable to provide the 95% confidence intervals in your datasets.
The remainder of my comments are as follows and from the order of appearance within the manuscript.
Line 47… probably around 30 million heads (Faye 2015)

Line 62-63 “The selection of the sampling method depends on disease progression (Rossi et al. 2018).”
I suggest rewording here. “Progression” implies the same disease over time. Whereas TW is more suited to infectious that samples assembled cells that are swept up from the lower airways, BAL is for diseases within the respiratory tract beyond the carina, and presumably generalized (asthma, environmental insult..)

Line 68-70 “BAL and TW samples are also useful in giving an insight on the severity and stage of inflammatory reactions in the lung and also in the detection of subclinical pulmonary diseases (Beech 1975; Caldow 2001).
Beech, 1975, did not describe BAL- it was not even commonly used in veterinary medicine until after the 1980’s. Please use an alternate reference and remove Beech.

80: Animals and experimental design
Please include the weights (if recorded) and sex distribution of your healthy and diseased cohorts: this will aid in the reader seeing how the groups compared phenotypically and by signalement.

Line 91 suggest changing this wording “ including clinical signs (like such as dyspnea…
Lines 92-94 Clinical scoring. Please describe who did the clinical scoring (one or several of the authors?) and whether the scoring was blinded (based on the material and methods this does not appear to be likely). As well, for a scoring scheme please provide a reference to peer reviewed work, and explain what the “modifications” to the scoring system were
Section : Bronchoscopy and collection of TW and BAL samples
This is a long section and can benefit by revising with reduction of the repetition of a number of sentences, as well as a number of language issues. The route for xylazine should be included (IV?). As well, when the TW was performed is appears as written to be bronchial (line 109-110). Was this the case? I found this part confusing, - how has the TW been performed to sample from mid cervical trachea, when the statement earlier indicates that the endoscope tip had passed the carina and this in the bronchial region. As written here- if fluid was indeed flushed into the bronchi, the authors will need to describe what parts of the respiratory tract epithelial lining was sampled, and whether this first procedure may have had an influence on the subsequent BAL cytology.
This should be clarified, as samples from beyond the carina can differ from cervical tracheal (more likely upper airway cells can blend into cervical tracheal mucous. This is also vital to clarify for others who would be repeating your work.

Line 150 “The observation of tracheal wall congestion” are the authors suggesting "hyperemia"?
I do not believe congestion can be determined on visual inspection- others (Koch et al. Endoscopic scoring of the tracheal septum in horses and its clinical relevance for) the evaluation of lower airway health in horses. Eq Vet J 2007) have attempted to identify this and no found relationship (it is subjective and subject to inter reader variation). I suggest removing this qualifier and place focus on the mucous scoring.

Line 169 … food material (singular)
Line 171 “contained significantly more cells (824 ± 401.1cel/μl) than”… should be cells/μl and throughout the rest of the manuscript and tables
Line 180- “epithelia cells (and elsewhere, should be “epithelial cells”, as is written on line 188.
Discussion
Lines 190-193- This portion is largely a repetition of your aims. It would be of greater interest to the reader for this to begin with your highlights of your key findings, ie, what were the most important findings from your work?

Lines 193-193 According to the methodology applied in this work this is not surprising. For TW to be used for culture, the optimal method is by transtracheal aspiration; insertion of a sterile flushing tube through a tracheal canula between tracheal rings below the larynx. Alternatively in the horse, a guarded sterile silastic tube within and outer plastic sleeve (Darien et al. A tracheoscopic technique for obtaining uncontaminated lower airway secretions for bacterial culture in the horse. Eq Vet J 1990) is now in common clinical use. I suggest the authors expand on this part as to alternatives to be considered for the camel if bacteriologic samples are to be collected.

Lines 196-197 “As there were no complications during the follow-up
examination of studied animals, we can confirm that BAL procedure is considered safe in camels”.

This statement of “no complications” lacks data on how the animals were monitored and followed up. In the horse it is not unusual to have transient fever one or 2 days following a BAL. Thus Thus in material and methods there needs to include additional information on how animals were monitored during and in particular after the sampling. To make this statement of “no complications” the reader should be informed as to what basis you can make this conclusion.

Lines 209-212
As noted above, the proportion (% of differentials) has a very closely associated codependent factor of other cell fluxes, particularly given that the cell numbers per set volume increased in the diseased cohort. Thus, it is important to include volume of fluid recovered in your material and methods, and results. Thereby in your discussion you can provide more robust comment which may reveal whether selected cell types (ie macrophages) actually decreased in total number, or simply this appeared to be due to an overall increase in neutrophils. As well, within the discussion it would be of great interest to the reader to have an expansion of the interpretation of the results, not solely “findings are consistent (or at odds) with others”. Such comment are observations, which would benefit from then follow with a discussion of potential interpretation/ significance of your findings. Where at odds, are there reasons for differing findings, are there weaknesses in the current study the authors need clarify? Thus a suggestion of decrease in macrophage proportion (BAL) or lymphocyte proportion (TW) - (lines 35-40 introduction) needs to be put into context: were these findings immunologically relevant decrease in these cell types, or simply a bystander effect of influx of neutrophils?
Line 238. Reference Riihimaki et al 2008, they performed endobronchial biopsy with cell immunohistochemical staining, and not tracheal wash sampling. Please remove this reference.

·

Basic reporting

No comment

Experimental design

No comment

Validity of the findings

No comment

Additional comments

Be consistent with spellings, e.g. epithelial cells (line 177 and others)

·

Basic reporting

There are many spelling and grammatical errors throughout the manuscript. The reviewer indicated some of them but the authors should read carefully the entire manuscript to correct all mistakes.

Experimental design

The study design is straightforward however, the diseased group should be better characterized in the results section. For example, the clinical score should be reported (mean +/- SD or median, IQR as appropriate) for both diseased and healthy camels. Also, what were the final or presumptive diagnoses based on the work-up performed at the hospital (e.g. viral, bacterial, fungal, or parasitic infections or non-infectious respiratory disease)? Pneumonia vs. upper airway infection? Bacterial vs. viral diseases? This information may be provided in table format.

Validity of the findings

Mast cell granules do not stain with Diff Quick (Leclere et al. J Vet Int Med 2006). How did you differentiate mast cells from other cells?

Additional comments

PeerJ #54751
Cytological analysis of tracheal wash and bronchoalveolar lavage fluid in dromedary camels
The authors report their findings on cytology of respiratory secretions collected in dromedary camels by tracheal wash (TW) and bronchoalveolar lavage (BAL). Animals sampled included 9 clinically healthy camels and 18 presenting signs of respiratory disease. Tracheal washes and BAL were performed trans-orally under endoscopy guidance in sedated camels. Samples were processed within 1 hour of collection, stained with Diff Quick and differential count obtained by counting 400 cells.
The main finding was that in healthy camels, neutrophil and epithelial cell proportions were higher in TW than BAL samples while lymphocyte proportions were lower in TW than in BAL. Camels with respiratory disease exhibited higher proportions of neutrophils in both TW and BAL compared to healthy camels.
According to the authors, this is the first report of cytological analysis of respiratory secretions collected by TW and BAL in dromedary camels. As such, these findings provide valuable information concerning normal airway cytology as well as abnormalities commonly identified in camels with clinical respiratory disease.

General comments:
This reviewer was able to find 2 manuscripts reporting TW cytology in dromedary camels during a cursory Google Scholar literature search:
- Habasha FG and Hussain MH. Cytological analysis of transtracheal washes from healthy camels in Al-diwaniya province. AL-Qadisiya Journal of Vet. Med. Sci.2014 (13) 1.
- Habasha FG. Clinical & Diagnostic Study of E-coli from Camels with pneumonia using VITEK 2 Compact, histopathology & conventional PCR. - qu.edu.iq
It does appear that this study is the first to report BAL cytology in camels and the authors should be congratulated for such an accomplishment.
It would be helpful in the introduction to be more about specific about the type of respiratory diseases responsible for high mortality rate (bacterial vs. viral infections? PI-3, pasteurellosis, etc.?). What respiratory diseases are the most commonly diagnosed in camels in Saudi Arabia?
The discussion of sampling of respiratory secretions in the introduction needs to be more nuanced. In horses, TW is preferred for diagnosis of respiratory infections (both cytology and microbiologic culture) whereas BAL is preferred for diagnosis of non-infectious pulmonary diseases. However in cattle, BAL was shown to be equivalent to TW for the diagnosis of common bacterial and viral respiratory pathogens (Doyle et al. J Vet Int Med 2017). Also, the authors should talk about the fact that respiratory cytology may provide a presumptive diagnosis of infection rapidly (Jocelyn et al. Equine Vet J 2018) but ultimately, identification of the etiologic agent requires bacterial or viral culture or PCR.
The study design is straightforward however, the diseased group should be better characterized in the results section. For example, the clinical score should be reported (mean +/- SD or median, IQR as appropriate) for both diseased and healthy camels. Also, what were the final or presumptive diagnoses based on the work-up performed at the hospital (e.g. viral, bacterial, fungal, or parasitic infections or non-infectious respiratory disease)? Pneumonia vs. upper airway infection? Bacterial vs. viral diseases? This information may be provided in table format.
Mast cell granules do not stain with Diff Quick (Leclere et al. J Vet Int Med 2006). How did you differentiate mast cells from other cells?
Presumably, some diseased camels were affected by bacterial pneumonia. In such cases, a large portion of BAL neutrophils would be expected to show degenerative changes and potentially intra- and extra-cellular bacteria. Please provide those findings in the results section and comment in the discussion.
Data regarding potential complications or side-effects related to TW/BAL observed during this study (or lack thereof) should be presented in the results section.
The discussion related to neutrophils should appear first after discussion of total cell count (e.g. line 209) and before discussion of macrophages, lymphocytes, etc. The reason is that the neutrophilia is the main cytological finding of importance. The other changes are related to the fact that if neutrophil proportion increases then, other cells will be present in smaller proportions (max 100%).
There are several spelling and grammatical errors throughout the manuscript. The reviewer indicated some of them but the authors should read carefully the entire manuscript to correct other mistakes.

Specific comments:
- Line 51: One article cited in the introduction in support of the statement that camels are susceptible to a variety of respiratory pathogens is not related to respiratory diseases and should be removed (Scaglione 2017).
- Line 92: It is preferable to say “… lung sounds and rate of breathing were recorded”.
- Line 96: consider writing “Camels with a clinical score less than 3 points were considered healthy while camels with a clinical of 3 points or higher were considered diseased.”
- Line 99: consider replacing “fixed” by “positioned”.
- Line 102: Replace “application of xylazine” by “injection of xylazine”. How was xylazine administered? Intramuscularly or intravenously?
- Line 103: spelling mistake: “mouth gauge” should be replaced by “mouth gag”.
- Line 104: consider “…to keep the mouth open to protect and allow easy passage of the endoscope”.
- Line 109: the TW was done via endoscopy using a catheter to instill and recover saline solution. Why did you advance the endoscope and catheter into bronchi (e.g. first of second generation) when you explain subsequently that you sampled mid-cervical trachea? Also, if saline is infused mid-cervical region, the fluid would flow to the lowest point of the trachea (thoracic inlet) and the tip of the catheter would have to be advanced further in order to aspirate it back. Please explain.
- Line 114: spelling mistake; replace “preformed” by “performed”.
- Line 117: correct “mouth gag”
- Line 123: spelling mistake; replace “installed” by “instilled”.
- Line 120: consider “As soon as the tube was wedged into the bronchus, the cuff…”
- Line 129: what is a Barker’s chamber? Total cell counts are usually determined using a hemocytomer.
- -line 135: Reference to Fig. 3 should be moved to the results section.
- Line 153: specify how many camels required use of the endoscope to pass the BAL tube.
- Line 157: cells labelled macrophages in Fig. 3 appear to be lysed and possess more neutrophil than macrophage morphological attributes. Because of the degree of cell lysis, these cell should not be identified. Another photomicrograph would be preferable.
- Line 167: “oral squamous epithelial cells”.
- Line 171: “cells/µl”
- Line 177: “epithelial cells…”
- Line 194: Others have perform trans-tracheal wash to bypass the upper airway thereby, decreasing the likelihood of oral contamination (see Habasha 2014). This possibility should be discussed.
- Line 239: Please discuss this finding of ciliocytophthoria as potential indicator of viral disease (Gelardi et al. 2016).

Figure 3: Indicate magnification or insert a scale for each photomicrograph (photo II is clearly at a lower magnification compared to the other photos.)
Figure 4: A scatter-plot with mean and SEM would be more appropriate to display the results and provide the range of values. Also, only indicate p-values < 0.05 on the graph.

---

## Round 0.2 · Minor Revisions

Thank you for addressing the reviewers' comments, however there are several minor changes requested as you will see from their reviews. Please address these requests.

·

Basic reporting

No comment

Experimental design

No comment

Validity of the findings

No comment

Additional comments

Much improved version. Some minor editorial comments, as below

Line 76-78 Sentence: “Although respiratory cytology may provide a presumptive diagnosis of infection rapidly but ultimately, identification of the etiologic agent requires bacterial or viral culture or PCR (Jocelyn et al. 2018)”
suggest rephrase to: “Although respiratory cytology may rapidly provide a presumptive diagnosis of infection, identification of the etiologic agent requires bacterial or viral culture or PCR (Jocelyn et al. 2018)’”

Line 107-110 Sentence “ Clinical score points were recorded by two authors, but not blindly, so it was not possible to ask the owner of the diseased camel about the case history and repeat the examination in the Veterinary Teaching Hospital. ” confusing, please reformulate.

Line 116 , 120 Dose of xylazine, one place uses0,1 mg/kg, the other 0.1 mg/kg. Please use one or the other for consistency

Line 117 “ a special mouth gag… (not gage)

Line 192 symbol… ± 39.2 cells/µlL)

Sentence line 212-215 The percentages of neutrophils in TW (78.83 ± 3.89 %) and BAL (31.42 ± 4.33%) samples from the camels with high rectal temperature were higher than their percentages in the TW (51.22 ± 5.9%) and BAL (14.76 ± 1.99%) samples from the animals with normal rectal temperature.
Discussion: lines 219-234. Much improved.
Line 235 … with saliva might have resulted from….

Line 285…..” cilia, which could have originated from the inflamed airways”
Lines 290-294 suggest rephrasing as
Finally, one of the limitations of the present study is the cell identification method. Although it is widely used for cytological analysis in several species (Jackson et al. 2013), staining with Diffquick may fail to reliably identify all cell types (Leclere et al. 2006). Specifically, the estimated frequency of mast cells in the BAL samples may have been affected by Diffquick staining

·

Basic reporting

The manuscript has been revised to the satisfaction of this reviewer.

Experimental design

The manuscript has been revised to the satisfaction of this reviewer.

Validity of the findings

The manuscript has been revised to the satisfaction of this reviewer.

Additional comments

Thank you for correcting the manuscript in response to this reviewer's comments. The only minor correction to make is to insert units Line 93 & 96 (e.g. years, kg)

---

## Round 0.3 · Minor Revisions

A couple more minor edits from a reviewer, and a question regarding your statement:

- "The higher frequency of neutrophils in the respiratory tract fluids (TW and BAL) from healthy camels, in comparison to other species, could be explained by their higher proportion in the peripheral blood of healthy camels (Hussen et al. 2019)."

Can you confirm that you are comparing the values between normal camels and other species, and if so, which species specifically? I can see the usefulness in comparing blood parameters between diseased animals of different species, especially here where normals may have not been established as well. However, in principle, I don't see the need in your manuscript to compare blood values between healthy members of different species.

Please address the reviewer and my comment and I look forward to your revised manuscript.

·

Basic reporting

Minior comments:
Line 121, please provide the route of the 0.05 mg/kg xylazine (I presume it was also IV as per induction?

Line 145, typo- "hemocytometer.

Experimental design

ok

Validity of the findings

ok

Additional comments

as above

·

Basic reporting

No comment

Experimental design

No Comment

Validity of the findings

No Comment

Additional comments

The authors have duly revised the manuscript.

---

## Round 0.4 · accepted · Accept

Thank you for updating your manuscript and addressing my query. I find your reporting acceptable for publication, best wishes in your research.

Sharif

ssjv